# Trajectories of medical care expenditure in the last year of life associated with long-term care utilization in frail older adults: A retrospective cohort study

Noriko Yoshiyuki[1☯], Takuma Ishihara[2☯], Ayumi Kono[3]*, Naomi Fukushima[1,3‡], Takeshi Miura[1‡], Katsunori Kaneko[4‡]

1 Department of Community-based Integrated Care science, School of Nursing, Osaka Metropolitan University, Osaka, Japan, 2 Innovative and Clinical Research Promotion Center, Gifu University Hospital, Gifu University, Gifu, Japan, 3 Long-term Care Insurance Division, Department of Health and Welfare, Izumi City Municipal, Osaka, Japan, 4 School of Economics, Osaka Metropolitan University, Osaka, Japan

☯ These authors contributed equally to this work.
‡ NF, TM and KK also contributed equally to this work.
* ayukono@omu.ac.jp

## Abstract

### Background

Medical care and long-term care utilization in the last year of life of frail older adults could be a key indicator of their quality of life. This study aimed to identify the medical care expenditure (MCE) trajectories in the last year of life of frail older adults by investigating the association between MCE and long-term care utilization in each trajectory.

### Methods

The retrospective cohort study of three municipalities in Japan included 405 decedents (median age at death, 85 years; 189 women [46.7%]) from a cohort of 1,658 frail older adults aged ≥65 years who were newly certified as support level in the long-term care insurance program from April 2012 to March 2013. This study used long-term care and medical insurance claim data from April 2012 to March 2017. The primary outcome was MCE over the 12 months preceding death. Group-based trajectory modeling was conducted to identify the MCE trajectories. A mixed-effect model was employed to examine the association between long-term care utilization and MCE in each trajectory.

### Results

Participants were stratified into four groups based on MCE trajectories over the 12 months preceding death as follows: rising (n = 159, 39.3%), persistently high (n = 143, 35.3%), minimal (n = 56, 13.8%), and descending (n = 47, 11.6%) groups. Home-based long-term care utilization was associated with increased MCE in the descending trajectory (coefficient, 1.48; 95% confidence interval [CI], 1.35–1.62). Facility-based long-term care utilization was associated with reduced MCE in the rising trajectory (coefficient, 0.59; 95% CI, 0.50–0.69).

**Data Availability Statement:** The data underlying the findings include potentially identifying participant information and cannot be shared publicly due to ethical/legal restrictions. However, researchers interested in collaborating with the SOHA principal investigator and assessing the data should contact the ethics committee, School of Nursing, Osaka Metropolitan University via email (gr-hab-rinri@omu.ac.jp).

**Funding:** This work was supported by the Japan Society for the Promotion of Science (grant numbers 17K19831 [2017-2018[ and 19K21595 [2019-2021] to A.K.). The funders had no role in the design of the study and collection, analysis, and interpretation of data and in writing the manuscript.

**Competing interests:** The authors declare that they have no competing interests.

**Abbreviations:** CI, confidence interval; IQR, interquartile range.

Both home-based (coefficient, 0.92; 95% CI, 0.85–0.99) and facility-based (coefficient; 0.53; 95% CI, 0.41–0.63) long-term care utilization were associated with reduced MCE in the persistently high trajectory.

## Conclusions

These findings may facilitate the integration of medical and long-term care models at the end of life in frail older adults.

## Introduction

Older adults require adequate end-of-life care [1] that includes concurrent medical treatment and long-term care due to a high risk of acute or chronic multimorbidity. Long-term care utilization is associated with different trajectories according to the cause of morbidity, including terminal illnesses such as cancer, organ failure such as congestive heart failure, chronic lung disease, chronic kidney disease; frailty, and sudden death [2–5]. Indeed, 36.6% of ambulatory frail older adults [6] died within 5 years of being certified as those who need support in the public long-term care insurance program of Japan, the country with the highest life expectancy worldwide (men: 81.41 and women: 87.45 years in 2019) [7]. Therefore, comprehensive end-of-life care, including long-term care and medical care utilization, is necessary for individualized provision of multidimensional care needs in clinical practice to sustain healthcare resources and optimize public health strategies in aging societies.

Many member countries of the Organization for Economic Co-operation and Development face challenges in managing medical care and long-term care budgets. The cost of medical and long-term care in these countries remains a significant burden [8]. Medical care expenditure generally increases at the end of life [9]. However, it is plausible that several medical care expenditure trajectories [10, 11] exist for older adults because the pattern of disabilities and medical-care-utilization trajectories may differ among individuals before death [12]. End-of-life medical care expenditure is characterized by chronic conditions [13, 14]. Several studies [11, 15, 16] have reported that long-term care utilization might reduce medical care expenditure or utilization. A proportion of older adults may have lower medical care expenditure owing to long-term care utilization in the last year of life.

Considering the growing interest in the relationship between medical and long-term care expenditure, this study analyzed the medical and long-term care insurance claims data collected electronically by the local governments for the Southern Osaka Health and Aging (SOHA) study [6]. Japan employs a public long-term care insurance program for older adults managed by the local municipal governments. The program has been operational since 2000 and covers home-based and facility-based long-term care [17] for older adults with long-term care needs, including ambulatory frail, chair-bound, and bed-bound individuals. In particular, SOHA has collected the claims data of ambulatory frail older adults who required support in public long-term care utilization for 10 years since 2012, and the present study intended to use their 5-year data from April 2012 to March 2017. Medical care expenditure trajectories in the last year of frail older adults have not yet been investigated. We hypothesized that the association between medical care expenditure and long-term care utilization would differ in each trajectory.

The present study aimed to identify the medical care expenditure trajectories in both outpatient clinics and inpatient settings in the last year of life of frail older adults using the claims data of deceased individuals by investigating the association between medical care expenditure

and long-term care utilization, which included home-based or facility-based care in each trajectory.

## Methods

### Study design and data collection

This retrospective study included decedents selected from the SOHA study [6] of 1,658 population-based frail older adults aged ≥65 years with newly certified support levels in the long-term care insurance program from April 2012 to March 2013.

 We selected a primary cohort of decedents from the SOHA study participants who were alive for at least 1 year between April 2012 and March 2017. Of the 560 decedents in the SOHA study during the 5-year follow-up period, 405 were included in the primary cohort, and 155 who died during the first 11 months of follow-up were excluded (Fig 1). Data for this study comprised three datasets, including long-term care, medical insurance claims, and resident registers, collected from electronic records obtained by each local government. The data were recorded monthly. After unique numbers were assigned to each insured individual to enable identification across data, all data were anonymized. We accessed the data for research purposes on March 28, 2018.

 As the data recording system for medical insurance claims data for individuals covered by public assistance is independent of medical insurance data, we could not obtain the medical insurance claims data.

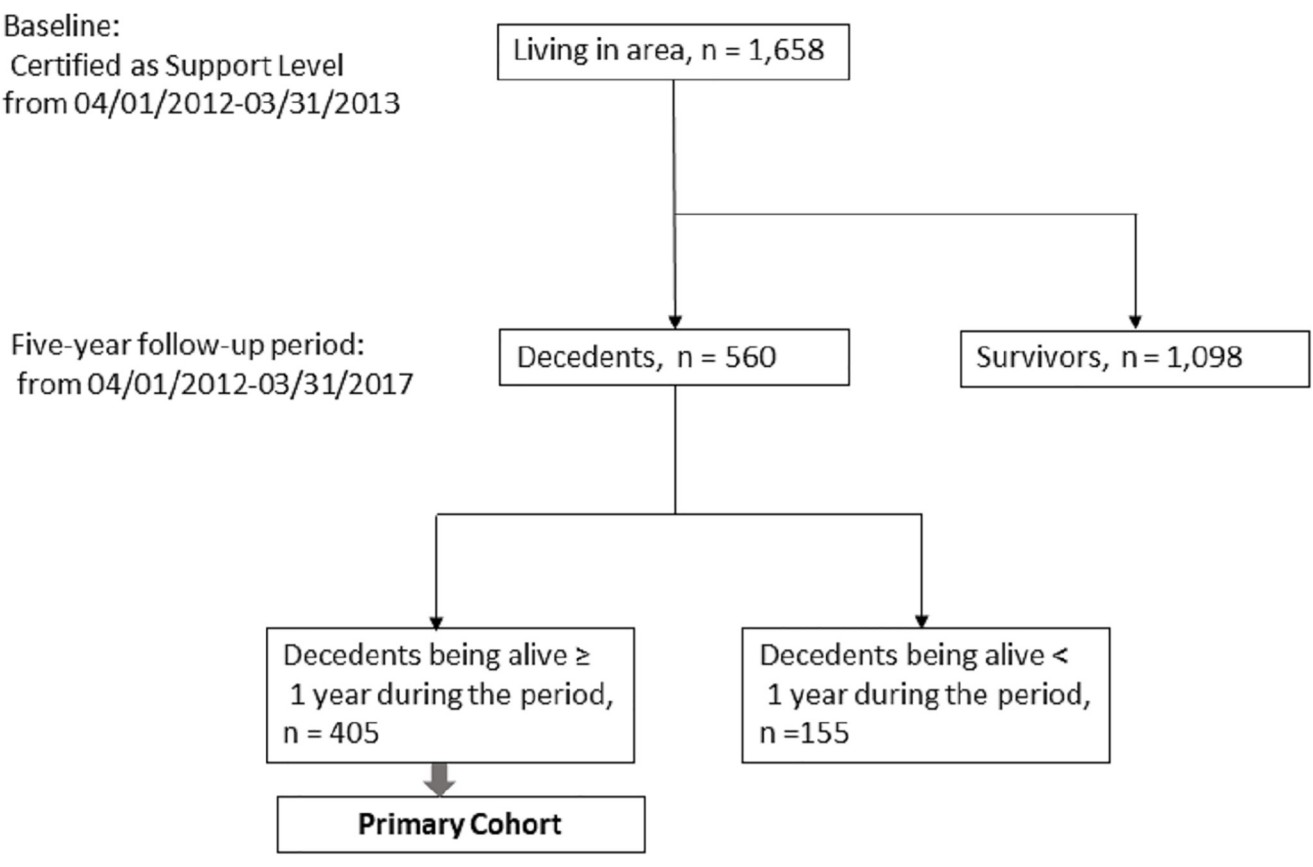

**Fig 1. Flow chart of selection of the primary cohort.**

## Long-term and medical care insurance system for older adults in Japan

The study setting was three municipalities located in the southern part of Osaka, Japan. The average proportion of older adults aged 65 years and over in these municipalities was 24.3%, similar to the national average of 24.1%, as of March 2013.

Under the Japanese long-term care insurance system, individuals aged ≥65 years who receive a certification of care need are eligible to use long-term care services. The care-need certification is categorized into seven levels, from the lowest to the highest care need (i.e., support levels 1–2 and care levels 1–5), based on the results of a national standard assessment of activities of daily living. Japan achieved universal medical insurance coverage in 1961, except for those receiving public assistance. The Late Elders' Medical Insurance Program was established in 2008 to provide medical insurance for all older adults aged ≥75 years.

## Outcome measures

The primary outcome of interest was monthly medical care expenditure in the 12 months before death, including inpatient/outpatient medical care payments and medication costs. Medical care expenditure was obtained from the healthcare insurance claims dataset. Expenditures are reported in U.S. dollars (the currency exchange rate for March 31, 2017, was 111.42 JPY per 1 U.S. dollars).

## Main exposures

Monthly long-term care utilization records were obtained from the long-term care insurance claims dataset. Home-based long-term care was defined as home-visit services, such as home aid, visiting nurses, visiting bathers, and visiting rehabilitation, and day and short-stay services. Facility-based long-term care was defined as any of the following three institutional care services covered by long-term care insurance: welfare, health, and medical facilities [17].

## Individual characteristics and covariates

Data on sex and age at death were collected from the resident register dataset. Long-term care expenditure in the last year of life was calculated by adding the monthly long-term care expenditures obtained from the claims dataset.

History of hospitalizations and chronic diseases were considered to account for health status in the last year of life. Five chronic disease groups were selected (i.e., cardiac disease, chronic respiratory disease, cancer, stroke, and dementia), which are the leading causes of death among frail adults [18, 19]. Five diseases were assigned by the World Health Organization International Classification of Diseases, Version 10 codes. Cardiac disease included chronic rheumatic heart diseases (I05-I09), ischemic heart diseases (I20-I25), pulmonary heart disease, diseases of pulmonary circulation (I26-I28), and other forms of heart disease (I30-I52). Chronic respiratory disease included chronic lower respiratory diseases (J40-J47), lung disease due to external agents (J60-J70), other respiratory diseases principally affecting the interstitium (J80-J84), suppurative and necrotic conditions of the lower respiratory tract (J85-J86), and other diseases of pleura (J90-J95). Cancer referred to malignant neoplasms (C00-C96), while stroke referred to cerebrovascular diseases (I60-I69). Dementia included vascular dementia (F01), dementia in other diseases classified elsewhere (F02), unspecified dementia (F03), and Alzheimer's disease (G30).

History of hospitalizations and International Classification of Diseases version 10 codes were obtained from the healthcare insurance claims dataset.

## Statistical analysis

Group-based trajectory modelling was performed using the SAS PROC TRAJ procedure to identify the distinct trajectories of medical care expenditure over the 12 months preceding death. We divided medical care expenditure into quartile-based categories. Decedents were assigned to one of the identified trajectories by calculating their membership probability in each latent cluster using a censored normal mixture model [20]. Akaike's information criterion and Bayesian information criterion were used to determine the number of trajectories. The order of each trajectory was fixed to three. As decedents who were on public assistance or used publicly funded medical care during the period may have been included, a sensitivity analysis was performed by excluding those who had not used medical care for more than two months immediately prior to death.

The characteristics of the decedents one year before death by trajectory group according to medical care expenditure were compared using the Fisher's exact test and Kruskal–Wallis test for discrete and continuous variables, respectively. Data are presented as the median (interquartile range [IQR]) for continuous variables or frequencies and percentages for discrete variables.

To examine the association between long-term care utilization and medical care expenditure as a continuous variable in each trajectory group, a linear mixed-effects model with participants as random effects and a correlation matrix with compound symmetry as the correlation structure among repeated measures were employed. The linear mixed-effects model included age, sex, hospitalization, chronic diseases, and months till death as fixed effects. The medical care expenditure value was naturally log-transformed to provide normality of the regression residuals. Coefficients obtained from the linear mixed model were back-transformed to indicate the percentage increase in medical care expenditure.

The sample size was based on data availability. The imputation method for missing data was not employed because no missing values of medical care expenditure were identified for ICD10 data. Analyses were conducted using SAS version 9.4 (SAS Institute Inc, Cary, NC) and R version 4.0.3 (R Foundation for Statistical Computing, Vienna, Austria). No adjustments for multiple comparisons were made. A two-sided $P$-value $< .05$ was considered significant.

## Ethical consideration

This study was approved by the Institutional Review Board on Human Subjects Research and Ethical Committee of the Graduate School of Nursing, Osaka City University, Japan (approval number: 29-6-1 and date of approval: February 2, 2018), and was conducted in accordance with the Japanese epidemiological guidelines of the Japanese Ministry of Health, Labor and Welfare and Ministry of Education, Culture, Sports and Technology. Informed consent from participants was obtained in the form of an opt-out method on the website of the Department of Community-based Integrated Care Science, School of Nursing, Osaka Metropolitan University (Osaka City University renamed Osaka Metropolitan University in 2022) for those who refused to participate in the study.

## Results

### Characteristics of study participants

The characteristics of the decedents included in this study are presented in Table 1. The cohort comprised 189 (46.7%) women and 216 (53.3%) men. The median age was 85 (IQR = 80–89) years. The median medical care expenditure for the last year of life was $20,271 (IQR = $4,955–$43,831). In total, 43 decedents (10.6%) used facility-based long-term care and 279 (68.9%) used home-based long-term care at least once during the last year of life.

**Table 1. Characteristics of decedents (N = 405).**

| Characteristics | Data |
|---|---|
| Sex, n(%) | |
| Male, n(%) | 216 (53.3) |
| Female, n (%) | 189 (46.7) |
| Age of death, years, median (IQR) | 85 (80–89) |
| Follow-up time[a], years, median (IQR) | 3 (2–4) |
| LTC expenditure in the last year of life, USD, median (IQR)[b] | 3,905 (611–11,146) |
| MCE in the last year of life, USD, median (IQR)[b] | 20,271 (4,955–43,831) |
| Chronic diseases[c], n (%) | |
| Cardiac disease | 268 (66.2) |
| Chronic respiratory disease | 246 (60.7) |
| Cancer | 124 (30.6) |
| Stroke | 157 (38.8) |
| Dementia | 94 (23.2) |
| Hospitalization [d], n (%) | 305 (75.3) |
| LTC utilization [e], n (%) | |
| Facility-based LTC | 43 (10.6) |
| Home-based LTC | 279 (68.9) |

IQR, interquartile range; MCE, medical care expenditure; LTC, long-term care.

[a] Years from LTC care-need certification to death.

[b] Currency exchange rate was 111.42 JPY per 1 USD as of March 31, 2017.

[c] Included decedents who were provided medical care for chronic diseases at least once during the last year of life.

[d] Included decedents who were hospitalized at least once during the last year of life.

[e] Included decedents who utilized LTC services at least once during the last year of life.

## Medical care expenditure trajectories in the last year of life

Parameter estimates of the trajectory analysis revealed that the three-trajectory model was better (Bayesian information criterion = −6652.45; Akaike's information criterion = −6622.42) than the four- (Bayesian information criterion = −6469.63, Akaike's information criterion = −6429.59) or five-trajectory models (Bayesian information criterion = −6382.92, Akaike's information criterion = −6332.88). In the three-trajectory model, a unique trajectory (i.e., descending medical care expenditure group) depicted by the four-trajectory model was eliminated. The characteristics of each trajectory were less distinct than those of the four-trajectory model. Thus, we selected the four-trajectory model comprising minimal (n = 56, 13.8%), descending (n = 47, 11.6%), rising (n = 159, 39.3%), and persistently high (n = 143, 35.3%) medical care expenditure for further analysis (Fig 2).

Over the course of the year, medical care expenditure in the persistently high medical care expenditure group was continually high and gradually increased. In contrast, the rising medical care expenditure group was persistently low until 6 months before death. This trend was similar in the descending medical care expenditure group, in which medical care expenditure accelerated particularly from 3 months before death. The trajectory of the descending medical care expenditure group diverged from that of the rising medical care expenditure group 6 months before death and decreased gradually. Medical care expenditure in the minimal group remained the lowest over the year.

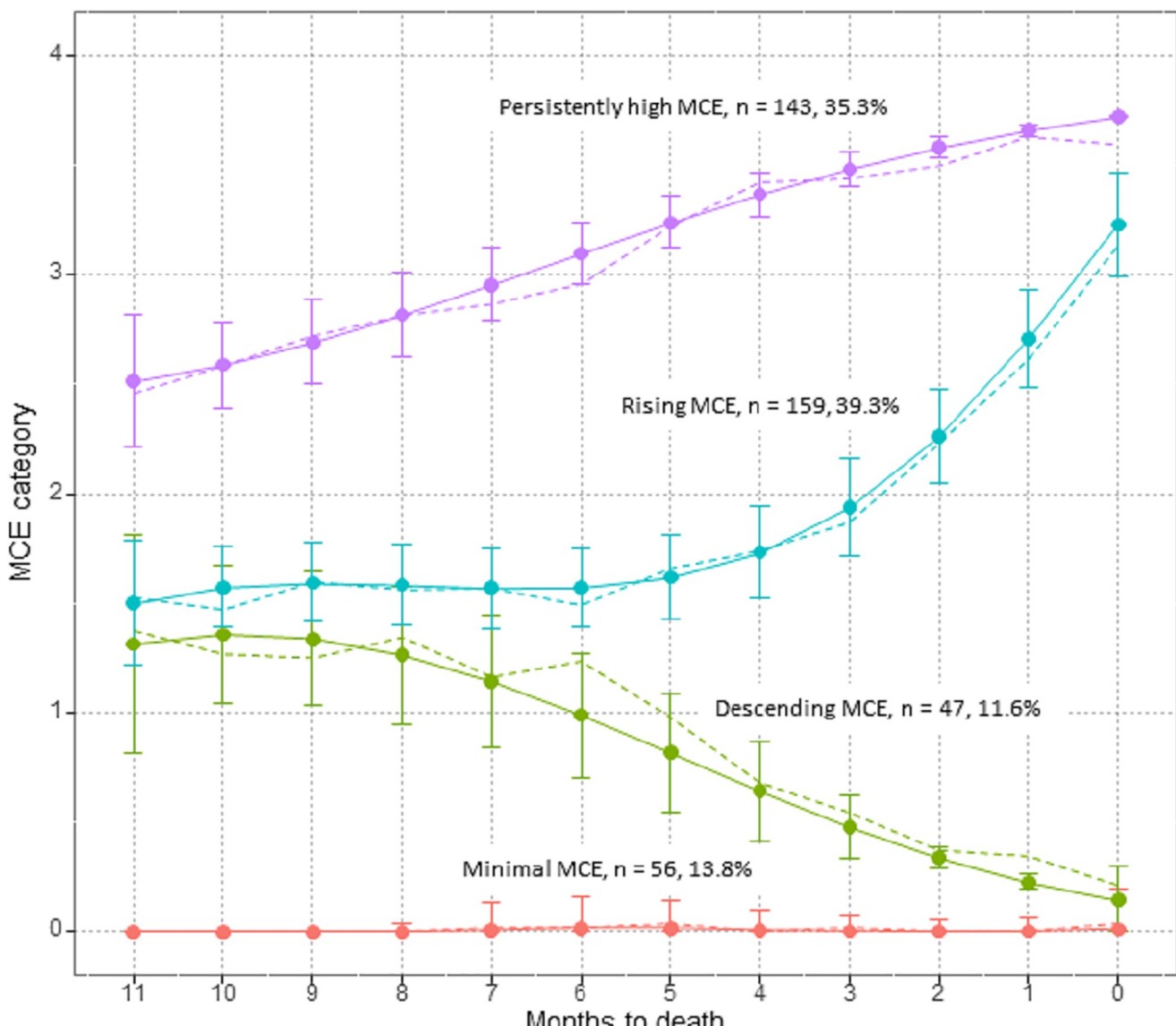

**Fig 2. Trajectories of medical care expenditure in the last year of life (N = 405).** MCE, medical care expenditure. MCE is categorized by quartiles of the median in the last year of life among decedents. The dashed lines indicate the observed mean MCE trajectories, solid lines indicate the predicted MCE trajectories, and error bars indicate the 95% confidence interval.

### Decedent characteristics according to medical care expenditure trajectory

The decedent characteristics according to medical care expenditure trajectory are presented in Table 2. The median age ranged from 82 years in the minimal medical care expenditure group to 87 years in the descending medical care expenditure group. Most decedents in the persistently high and rising medical care expenditure groups were hospitalized at least once during the last year of life (Table 3).

In the last year of life, the highest medical care expenditure was observed in the persistently high medical care expenditure group (median, $49,022; IQR, $39,584–$64,806), followed by the rising medical care expenditure group (median, $14,217; IQR, $8,533–$22,751; Table 2).

**Table 2. Characteristics of decedents according to medical care expenditure trajectories (N = 405).**

| Characteristics | N (%) | | | | p-value |
|---|---|---|---|---|---|
| | Minimal MCE group (n = 56) | Descending MCE group (n = 47) | Rising MCE group (n = 159) | Persistently high MCE group (n = 143) | |
| Female, n (%) | 27 (48.2) | 26 (55.3) | 71(44.7) | 65 (45.5) | 0.62 |
| Age at death, years, median (IQR) | 82 (73–87) | 87 (86–91) | 86 (83–90) | 83 (79–88) | < .001 |
| Follow-up time[a], years, median (IQR) | 3 (2–4) | 5 (4–5) | 3 (2–4) | 3 (2–4) | < .001 |
| LTC expenditure in the last year of life, USD[b], median (IQR) | 5,969 (540–17,043) | 8,077 (1,003–17,099) | 5,116 (2,352–12,017) | 2,070 (0–5,788) | < .001 |
| Total MCE in the last year of life, USD[b], median (IQR) | 0 (0–0) | 2,909 (1,699–10,300) | 14,217 (8,533–22,751) | 49,022 (39,584–64,806) | < .001 |
| Chronic diseases[c], n (%) | | | | | |
| Cardiac disease | 1 (1.8) | 29 (61.7) | 127 (79.9) | 111 (77.6) | < .001 |
| Chronic respiratory disease | 1 (1.8) | 21 (44.7) | 113 (71.1) | 111 (77.6) | < .001 |
| Cancer | 0 (0) | 9 (19.1) | 55 (34.6) | 60 (42.0) | < .001 |
| Stroke | 1 (1.8) | 16 (34.0) | 65 (40.9) | 35 (24.5) | < .001 |
| Dementia | 1 (1.8) | 10 (21.3) | 48 (30.2) | 44 (28.9) | < .001 |
| Hospitalization[d], n (%) | 1 (1.8) | 18 (38.3) | 144 (90.6) | 142 (99.3) | < .001 |
| LTC utilization[e], n (%) | | | | | |
| Facility-based LTC | 8 (14.3) | 14 (29.8) | 13 (8.2) | 8 (5.6) | < .001 |
| Home-based LTC | 33 (58.9) | 30 (63.8) | 124 (78.0) | 92 (64.3) | .01 |

Abbreviations: IQR, interquartile range; MCE, medical care expenditure; LTC, long-term care.

[a] Years from LTC care-need certification to death.

[b] Currency exchange rate was 111.42 JPY per 1 USD as of March 31, 2017.

[c] Included decedents who were provided medical care for the chronic diseases at least once during the last year of life.

[d] Included decedents who were hospitalized at least once during the last year of life.

[e] Included decedents who utilized LTC services at least once during the last year of life.

**Table 3. Hospitalization during each month in the last year of life according to medical care expenditure trajectory.**

| Months until death | Total (N = 405) | MCE trajectories | | | |
|---|---|---|---|---|---|
| | | Minimal group (n = 56) | Descending group (n = 47) | Rising group (n = 159) | Persistently high group (n = 143) |
| 11 | 65 (16.0) | 0 (0) | 6 (12.8) | 13 (8.2) | 46 (32.2) |
| 10 | 63 (15.6) | 0 (0) | 5 (10.6) | 10 (6.3) | 48 (33.6) |
| 9 | 81 (20.0) | 0 (0) | 5 (10.6) | 14 (8.8) | 62 (43.4) |
| 8 | 84 (20.7) | 0 (0) | 7 (14.9) | 8 (5.0) | 69 (48.3) |
| 7 | 95 (23.5) | 0 (0) | 7 (14.9) | 14 (8.8) | 74 (51.7) |
| 6 | 94 (23.2) | 0 (0) | 9 (19.1) | 12 (7.5) | 73 (51.0) |
| 5 | 113 (27.9) | 0 (0) | 5 (10.6) | 14 (8.8) | 94 (65.7) |
| 4 | 125 (30.9) | 0 (0) | 4 (8.5) | 18 (11.3) | 103 (72.0) |
| 3 | 138 (34.1) | 0 (0) | 2 (4.3) | 27 (17.0) | 109 (76.2) |
| 2 | 172 (42.5) | 0 (0) | 1 (2.1) | 51 (32.1) | 120 (83.9) |
| 1 | 204 (50.4) | 0 (0) | 0 (0) | 75 (47.2) | 129 (90.2) |
| 0 | 250 (61.7) | 1 (1.8) | 1 (2.1) | 119 (74.8) | 129 (90.2) |

MCE, Medical care expenditure.

Regarding the incidence of chronic diseases, cardiac and chronic respiratory diseases were the two most common diseases among decedents in the three medical care expenditure groups, except in the minimal medical care expenditure group. The proportion of cancer was relatively high among decedents in the rising (34.6%) and persistently high (40.2%) medical care expenditure groups, and the proportion of stroke was relatively high among decedents in the descending (34%) and rising (40.9%) medical care expenditure groups.

## Long-term care utilization in the last year of life according to medical care expenditure trajectories

In the last year of life, the highest long-term care expenditure was observed for decedents in the descending medical care expenditure group (median, $8,077; IQR, $10,003–$17,099), whereas the lowest one was observed for those in the persistently high medical care expenditure group (median, $2,070; IQR, $0–$5,788) (Table 2).

The frequency distribution of long-term care utilization during each month in the last year of life according to the medical care expenditure trajectory is depicted in Fig 3. Over the course of the year, the number of decedents who used facility-based long-term care did not change substantially. Decedents who utilized facility-based long-term care in the descending medical care expenditure group comprised the largest number (Fig 3A; range: 11–17%), followed by the minimal medical care expenditure group (Fig 3B; range: 11–13%). Very few decedents in the rising (Fig 3C) and persistently high (Fig 3D) medical care expenditure groups used

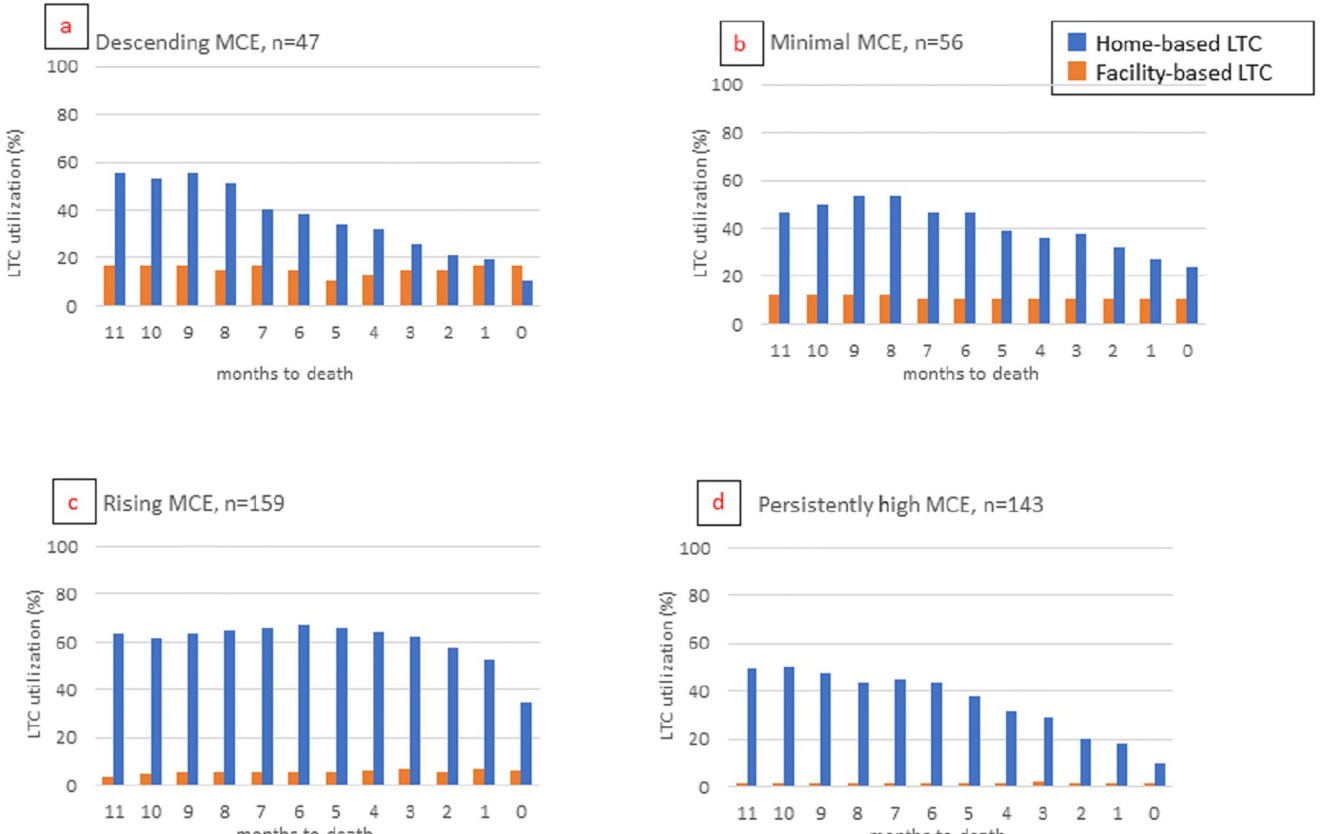

**Fig 3. Long-term care utilization in the last year of life according to medical care expenditure trajectories.** LTC, long-term care; MCE, medical care expenditure.

**Table 4. Associations between long-term care utilization and medical care expenditure according to medical care expenditure trajectories (N = 405).**

|  | Minimal MCE | Descending MCE | Rising MCE | Persistently high MCE |
|---|---|---|---|---|
| Facility-based LTC |  |  |  |  |
| Coeff (95% CI) | 1.01 (1.00–1.02) | 1.04 (0.93–1.20) | 0.59 (0.50–0.69) | 0.53 (0.41–0.63) |
| *p*-value | .28 | .53 | < .001 | < .001 |
| Home-based LTC |  |  |  |  |
| Coeff (95% CI) | 1.0 (0.99–1.00) | 1.48 (1.35–1.62) | 0.95 (0.89–1.02) | 0.92 (0.85–0.99) |
| *p*-value | .27 | < .001 | .16 | .03 |

CI, confidence interval; Coeff, coefficient; LTC, long-term care; MCE, Medical care expenditure.

Mixed-effect models were adjusted for sex, age, hospitalization, chronic diseases, and months to death. The model of the minimal medical care expenditure group was unadjusted for chronic diseases owing to the low number of decedents who were provided health care for chronic diseases.

Medical care expenditure was log-transformed.

facility-based long-term care. The number of decedents who used home-based long-term care was similar and decreased gradually in three of the four medical care expenditure groups, whereas >50% of decedents in the rising medical care expenditure group used home-based long-term care each month until a month before death.

The mixed-effects model adjusted for sex, age, hospitalization, chronic diseases, and time indicated that home-based long-term care utilization increased medical care expenditure in the descending medical care expenditure group (coefficient, 1.48; 95% confidence interval (CI), 1.35–1.62; $P < .001$). Facility-based long-term care utilization reduced medical care expenditure in the rising medical care expenditure group (coefficient, 0.59; 95% CI, 0.50–0.69; $P < .001$). Facility-based (coefficient, 0.53; 95% CI, 0.41–0.63; $P < .001$) and home-based (coefficient, 0.92; 95% CI, 0.85–0.99; $P = .03$) long-term care utilization reduced medical care expenditure in the persistently high medical care expenditure group (Table 4).

## Sensitivity analysis regarding trajectories of medical care expenditure in the last year of life

The sensitivity analysis results for trajectories of medical care expenditure in the last year of life excluding decedents who had not used medical care for more than two months prior to death are presented in S1 Fig. Persistently middle medical care expenditure (n = 33, 9.3%) was estimated instead of minimal medical care expenditure. The other three trajectories—persistently high (n = 140, 40%), rising (n = 147, 42.0%), and descending medical care expenditure (n = 34, 9.6%)—were estimated.

## Discussion

In this retrospective cohort study of decedents, four distinct medical care expenditure trajectories were identified in the last year of life of frail older adults. Significant differences were observed in the association between medical care expenditure and long-term care utilization among medical care expenditure trajectories based on a mixed-effects model.

This study found that 39.3% of the decedents were in the rising medical care expenditure trajectory, in which medical care expenditure increased sharply during the last 6 months before death. Moreover, 35.3% had a persistently high medical care expenditure trajectory, which exhibited a gradually increasing trend, and 90.2% of those were hospitalized a month before death. Furthermore, 74.6% had an increasing medical care expenditure in the last year of life, similar to the previous study that reported that medical care expenditure increased in

the last year [21]. In contrast, 11.6% of the decedents had a descending medical care expenditure trajectory concomitant with decreasing medical care expenditure during the last 6 months of life.

Regarding each medical care expenditure trajectory in the last year of life of frail older adults, the association between long-term care utilization and medical care expenditure differed in the following ways.

Decedents in the rising medical care expenditure who were more likely to use facility-based long-term care had lower medical care expenditure (0.59-fold) than did those who were less likely to use it in the present study. They were more likely to experience a stroke (40.9%) with progressive functional decline. Previous studies [3, 4, 12] have reported a functional decline in certain decedent groups within 6 months preceding death. These decedents could utilize both medical care and home-based long-term care more frequently than facility-based long-term care.

Decedents in the persistently high medical care expenditure trajectory who were more likely to utilize long-term care had lower medical care expenditure than those who were less likely to utilize it (home-based long-term care = 0.92-fold, facility-based long-term care = 0.53-fold). They might have employed intensive medical care at the end of life instead of long-term care because they were more likely to have cancer (42.0%) than were those in other trajectories.

For decedents in both the rising and persistently high medical care expenditure trajectories, a lower incidence ratio of medical care expenditure was associated with the use of facility-based long-term care. Of note, one of the institutions that provide long-term care is a medical facility and offer a certain degree of medical or nursing care, leading to a reduction in hospital-based medical care expenditure.

In the descending medical care expenditure trajectory, decedents who were more likely to use home-based long-term care had higher medical care expenditure than those who were less likely to utilize it (1.48-fold), and 11–17% of the decedents were institutionalized in the last year of life. Older adults at the end of life need palliative care to relieve suffering, and home-based long-term care might increase home-based medical care through visits by physicians or nurses. However, their yearly median total expenditure on medical and long-term care ($10,986) was significantly lower than those of the rising ($19,333) or persistently high ($51,092) medical care expenditure groups. Furthermore, these decedents survived longer (age at death: 88 years; years from care-need certification: 5); therefore, they might have adopted the most favorable care utilization.

Furthermore, the degree of home-based long-term care utilization of decedents in the minimal medical care expenditure trajectory was similar to that in the descending trajectory, and 11–13% of the decedents in the minimal group were institutionalized in the last year of life. They might comprise those who died suddenly without medical care or those who lacked continuous records of medical care utilization in the claims dataset due to publicly funded medical care (e.g., for intractable diseases or disabilities of veterans) or public assistance. As only one decedent had a chronic disease per chronic disease type, they may have died suddenly without requiring medical care.

According to the OECD Health Statistics 2019 [22], long-term care expenditure constituted 1.8% of GDP in Japan, which is a similar level to that of the OECD average of 1.7% in 2017. Given the growth in aging populations worldwide, the demand for long-term care at the end-of-life is increasing rapidly [23, 24]. Age has been reported to have the largest effect on the increase in medical and long-term care expenses [8, 11, 25–27] and may play an essential role in predicting the relationship between medical and long-term care.

This study has some limitations. First, although we prevented a selection bias by analyzing all medical care expenditure trajectories, including the minimal one, the dataset did not

provide precise information on medical care expenditure for decedents who used either publicly funded medical care or required public assistance in all trajectories during follow-up. Nevertheless, we excluded decedents who received public assistance owing to unavailability of data on their newly certified support level from April 2012 to March 2013. Our approach may have overlooked those whose medical care was covered by medical care insurance at the start of the follow-up but was shifted to public assistance at later time points. Particularly, decedents in the minimal medical care expenditure group might comprise those who died suddenly without medical care or those who lacked continuous records of medical care utilization in the claims dataset, and we could not distinguish them. The claims data for publicly funded medical care or public assistance are managed separately in Japan's healthcare system, and these claims datasets should be merged in further investigation.

Second, the International Classification of Diseases version 10 scores did not fully assess chronic diseases, which may contribute substantially to medical care expenditure. In the present study, the International Classification of Diseases version 10 was limited to the diagnosis of individuals who accessed medical care during the follow-up period. Individuals with chronic diseases who did not access a hospital or clinic were excluded from the chronic disease data.

Third, our study design could not determine the causal factors of the medical care expenditure trajectory patterns, although frailty could predict higher expenditure [28].

Our information was based only on the claims dataset and was not linked to the detailed physical and psychosocial aspects of the decedents. Thus, future studies are warranted to clarify the quality of end-of-life care of individuals in each medical care expenditure trajectory by analyzing data on subjective information, including functional independence or satisfaction of older adults or their bereaved family members.

This study provides novel information regarding the association between medical care expenditure and long-term care utilization in the last year of life in frail older adults. Four medical care expenditure trajectories were characterized according to long-term care utilization. Knowledge about the course of medical care expenditure could enhance the integration of medical and long-term care management models for dignified end of life in frail older adults.

## Supporting information

**S1 Fig. Sensitivity analysis regarding trajectories of medical care expenditure in the last year of life excluding decedents who had not used medical care for more than two months prior to death (N = 354).**
(TIF)

## Acknowledgments

The authors would like to thank the older adults and their family members for participating in this study. We express our gratitude to the staff of the Long-term Care Insurance and the Health Care Insurance Sections of Izumi, Izumiotsu, and Misaki Local Government Offices, Osaka Federation of National Health Insurance Organization, and Osaka Prefecture Association of Medical Care Services for Older Senior Citizens.

## Author Contributions

**Conceptualization:** Noriko Yoshiyuki, Ayumi Kono.

**Data curation:** Noriko Yoshiyuki, Takuma Ishihara, Ayumi Kono, Naomi Fukushima.

**Formal analysis:** Takuma Ishihara.

**Funding acquisition:** Ayumi Kono.

**Investigation:** Noriko Yoshiyuki, Takuma Ishihara, Ayumi Kono.

**Methodology:** Noriko Yoshiyuki, Takuma Ishihara.

**Project administration:** Ayumi Kono, Naomi Fukushima.

**Resources:** Ayumi Kono.

**Software:** Takuma Ishihara.

**Supervision:** Ayumi Kono, Naomi Fukushima, Takeshi Miura, Katsunori Kaneko.

**Validation:** Noriko Yoshiyuki, Ayumi Kono, Naomi Fukushima.

**Visualization:** Noriko Yoshiyuki.

**Writing – original draft:** Noriko Yoshiyuki, Takuma Ishihara, Ayumi Kono.

**Writing – review & editing:** Noriko Yoshiyuki, Takuma Ishihara, Ayumi Kono, Naomi Fukushima, Takeshi Miura, Katsunori Kaneko.

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
