## [Decision Letter · Decision Letter 0]

25 Oct 2023

PONE-D-23-27675Trajectories of medical care expenditure in the last year of life associated with long-term care utilization in frail older adults: a retrospective cohort studyPLOS ONE

Dear Dr. Kono,

Thank you for submitting your manuscript to PLOS ONE. After careful consideration, we feel that it has merit but does not fully meet PLOS ONE’s publication criteria as it currently stands. Therefore, we invite you to submit a revised version of the manuscript that addresses the points raised during the review process.

We look forward to receiving your revised manuscript.

Kind regards,

Ramzi Ibrahim, M.D.

Academic Editor

PLOS ONE

Journal Requirements:

Reviewers' comments:

Reviewer's Responses to Questions

**Comments to the Author**

1. Is the manuscript technically sound, and do the data support the conclusions?

Reviewer #1: Partly

2. Has the statistical analysis been performed appropriately and rigorously? 

Reviewer #1: Yes

3. Have the authors made all data underlying the findings in their manuscript fully available?

Reviewer #1: Yes

4. Is the manuscript presented in an intelligible fashion and written in standard English?

Reviewer #1: No

5. Review Comments to the Author

Reviewer #1: Thank you for letting me review this clinically important paper. I agree with the initial hypothesis that end-of-life medical care expenditure trajectories for older adults can be heterogenous. In a similar vein, within a study of elderly cancer patients on home-based palliative care, several differing trajectories of healthcare utilization were also seen. [https://doi.org/10.1186/s12916-022-02513-y]

These are some of my comments for consideration:

1) Long-term care is understood differently depending on country context. Suggest providing a few sentences at the Introduction section to set the stage (instead of leaving it till later within the Methods section)

2) The first paragraph talks about frailty but then frailty barely mentioned in the remaining article. Even in Table 1, there is no capture of clinical data with respect to frailty (e.g. function, frailty scale, comorbidity severity, cognition, etc). In my opinion, elderly age and presence of a chronic disease is not sufficient to characterize frailty. Suggest rewriting the first paragraph to focus on the significance of studying medical care expenditure in EOL among elderly adults and linking it subsequently to why there is a eventual hypothesis that LTC is associated with different trajectories of care expenditure.

The alternative is to include more clinical data/indicators for frailty, of which perhaps what would be also of interest would be frailty predictors for medical care expenditure.

3) It is interesting that within the Minimal MCE group, the total MCE in last year of life is 0 (0-0). Numbers are also not small (56/405 of cohort). Is this truly zero utilization of medical care? Or is this due to missing data (e.g. patient was recruited into the cohort but left the healthcare coverage area? Some patients did not file insurance claims?) If this is truly due to zero utilization, the authors could provide a statement to declare the comprehensive coverage of the primary outcome measure (healthcare insurance claims). Alternative is to consider a sensitivity analysis of trajectories excluding those patients with zero MCE or imputing their MCE.

- Moreover, very few patients (1 per chronic disease type) within these 56 had a known chronic disease. Hence, on deterioration (prior to death), one would expect MCE of some form for further investigations?

- I note that this has been subsequently mentioned in the discussion portion (about the lacking of continuous records of medical care utilization in the claim dataset). This missingness may need to be handled or at least written clearly as a limitation.

4) Suggest to reword as associations instead of causations. E.g. “home-based long-term care utilization was associated with increased medical care expenditure in the descending medical care expenditure group. Also I believe the testing of association between long-term care and MCE within group-based trajectory modelling is an incidence rate ratio which should be mentioned?

5) The paragraphs within discussion describing the possible associations with LTC and MCE trajectories are difficult to understand and need to be revised.

- For example, for those with rising/persistently high MCE trajectories, a lower incidence rate ratio of MCE is associated with use of facility-based long-term care. The hypothesized reason behind this association should be clearly spelled out. (could it be because facility-based LTC provide some degree of medical/nursing care that reduces the need for hospital MCE?)

- On another note, those with descending MCE trajectories, a higher incidence rate ratio of MCE is associated with use of home-based LTC. What is the hypothesized reason behind this association?

6) Under limitations, it is mentioned that ICD-10 did not fully assess chronic disease. However, a common methodology is to compute comorbidity severity (either charlson or elixhauser) via ICD-10 codes. This could be considered as a surrogate measure of comorbidity severity since the authors have access to ICD codes

7) The fourth limitation is unclear to me. May need to be rephrased.

8) Under conclusion, the first conclusion that there are heterogenous MCE trajectories is quite clear. The subsequent sentences are unclear and I am not sure what the authors are recommending. Are you suggesting that improving on long-term care utilization may attenuate high MCE?

6. PLOS authors have the option to publish the peer review history of their article (what does this mean?). If published, this will include your full peer review and any attached files.

Reviewer #1: No

---

## [Author Response · Author response to Decision Letter 0]

15 Nov 2023

Point-by-point responses to the reviewer’s comments

We appreciate the reviewer’s helpful suggestions. Our point-by-point responses to the reviewer’s comments are provided below. In addition to the changes suggested by the reviewer, we have revised some other sentences at various points to improve their flow and comprehensibility and reduce redundancy. The newly added/revised text has been highlighted yellow in the revised manuscript.

Reviewer 1

1) Long-term care is understood differently depending on country context. Suggest providing a few sentences at the Introduction section to set the stage (instead of leaving it till later within the Methods section)

Response

According to your suggestion, we have added a few sentences to the Introduction to set the stage for non-Japanese readers to understand the context of long-term care in Japan (L100-L107).

2) The first paragraph talks about frailty but then frailty barely mentioned in the remaining article. Even in Table 1, there is no capture of clinical data with respect to frailty (e.g. function, frailty scale, comorbidity severity, cognition, etc). In my opinion, elderly age and presence of a chronic disease is not sufficient to characterize frailty. Suggest rewriting the first paragraph to focus on the significance of studying medical care expenditure in EOL among elderly adults and linking it subsequently to why there is a eventual hypothesis that LTC is associated with different trajectories of care expenditure.

The alternative is to include more clinical data/indicators for frailty, of which perhaps what would be also of interest would be frailty predictors for medical care expenditure.

Response

As you suggested, we have rewritten the first paragraph to focus on the significance of studying medical care expenditure and long-term care utilization among older adults at the end of life(L73–L85).

3) It is interesting that within the Minimal MCE group, the total MCE in last year of life is 0 (0-0). Numbers are also not small (56/405 of cohort). Is this truly zero utilization of medical care? Or is this due to missing data (e.g. patient was recruited into the cohort but left the healthcare coverage area? Some patients did not file insurance claims?) If this is truly due to zero utilization, the authors could provide a statement to declare the comprehensive coverage of the primary outcome measure (healthcare insurance claims). Alternative is to consider a sensitivity analysis of trajectories excluding those patients with zero MCE or imputing their MCE.

Response

Decedents whose total MCE in the last year of life was zero were followed up until their death, meaning the zero MCE was not due to missing data. However, we could not distinguish decedents who did not use medical care genuinely from those who used publicly funded medical care or required public assistance during the follow-up period. We have mentioned this point in the limitations. We assumed that decedents who used publicly funded medical care or required public assistance had not used medical care for more than two months immediately prior to death. Therefore, as you suggested, we have added a sensitivity analysis to estimate the trajectory of medical care expenditure for this group (L200-L203) (L353-L361). 

- Moreover, very few patients (1 per chronic disease type) within these 56 had a known chronic disease. Hence, on deterioration (prior to death), one would expect MCE of some form for further investigations?

Response

As you have clearly noted, for each chronic disease type, only one decedent in the minimal MCE group had a chronic disease. These individuals may have died suddenly without medical care. We have added this explanation to the Discussion (L419–L421), as you suggested.

- I note that this has been subsequently mentioned in the discussion portion (about the lacking of continuous records of medical care utilization in the claim dataset). This missingness may need to be handled or at least written clearly as a limitation.

Response

As you suggested, we have clarified the lack of continuous records of medical care utilization in the claims dataset in the Limitations (L429–L443).

4) Suggest to reword as associations instead of causations. E.g. “home-based long-term care utilization was associated with increased medical care expenditure in the descending medical care expenditure group. Also I believe the testing of association between long-term care and MCE within group-based trajectory modelling is an incidence rate ratio which should be mentioned?

Response

Thank you for your suggestion. We have reworded the explanation as association to avoid causation (L402–L405).

5) The paragraphs within discussion describing the possible associations with LTC and MCE trajectories are difficult to understand and need to be revised.

- For example, for those with rising/persistently high MCE trajectories, a lower incidence rate ratio of MCE is associated with use of facility-based long-term care. The hypothesized reason behind this association should be clearly spelled out. (could it be because facility-based LTC provide some degree of medical/nursing care that reduces the need for hospital MCE?)

Response

Thank you for your helpful suggestion. We have added the explanation to the Discussion section (L396–L401).

- On another note, those with descending MCE trajectories, a higher incidence rate ratio of MCE is associated with use of home-based LTC. What is the hypothesized reason behind this association?

Response

Thank you for the question. Older adults at the end of life need palliative care, including medication, fluid replacement, or tube feeding, to relieve suffering. The patients receiving home-based long-term care might also access home-based medical care through visits by physicians or nurses. We have added this point to the Discussion (L405–L407).

6) Under limitations, it is mentioned that ICD-10 did not fully assess chronic disease. However, a common methodology is to compute comorbidity severity (either charlson or elixhauser) via ICD-10 codes. This could be considered as a surrogate measure of comorbidity severity since the authors have access to ICD codes

Response

We appreciate your suggestion. Our previous description of this limitation was insufficient. We have revised this limitation for clarity (L446–L449).

7) The fourth limitation is unclear to me. May need to be rephrased.

Response

We think this limitation is not important and have deleted it.

8) Under conclusion, the first conclusion that there are heterogenous MCE trajectories is quite clear. The subsequent sentences are unclear and I am not sure what the authors are recommending. Are you suggesting that improving on long-term care utilization may attenuate high MCE?

Response

As you suggested, we have revised the conclusion to clarify the recommendation (L462–L464).

---

## [Decision Letter · Decision Letter 1]

2 Jan 2024

Trajectories of medical care expenditure in the last year of life associated with long-term care utilization in frail older adults: a retrospective cohort study

PONE-D-23-27675R1

Dear Dr. Kono,

We’re pleased to inform you that your manuscript has been judged scientifically suitable for publication and will be formally accepted for publication once it meets all outstanding technical requirements.

Kind regards,

Ramzi Ibrahim, M.D.

Academic Editor

PLOS ONE

Additional Editor Comments (optional):

Reviewers' comments:

Reviewer's Responses to Questions

**Comments to the Author**

1. If the authors have adequately addressed your comments raised in a previous round of review and you feel that this manuscript is now acceptable for publication, you may indicate that here to bypass the “Comments to the Author” section, enter your conflict of interest statement in the “Confidential to Editor” section, and submit your "Accept" recommendation.

Reviewer #1: All comments have been addressed

2. Is the manuscript technically sound, and do the data support the conclusions?

Reviewer #1: Partly

3. Has the statistical analysis been performed appropriately and rigorously? 

Reviewer #1: Yes

4. Have the authors made all data underlying the findings in their manuscript fully available?

Reviewer #1: Yes

5. Is the manuscript presented in an intelligible fashion and written in standard English?

Reviewer #1: Yes

6. Review Comments to the Author

Reviewer #1: My queries have been addressed. No further questions.

7. PLOS authors have the option to publish the peer review history of their article (what does this mean?). If published, this will include your full peer review and any attached files.

Reviewer #1: No

---

## [Editor Report · Acceptance letter]

16 May 2024

PONE-D-23-27675R1 

PLOS ONE

Dear Dr. Kono, 

I'm pleased to inform you that your manuscript has been deemed suitable for publication in PLOS ONE. Congratulations! Your manuscript is now being handed over to our production team.

Kind regards, 

on behalf of

Dr. Ramzi Ibrahim 

Academic Editor

PLOS ONE